# Simplifying Node Classification on Heterophilous Graphs with Compatible Label Propagation

**Zhiqiang Zhong**                                              *zhiqiang.zhong@uni.lu*
*University of Luxembourg*

**Sergey Ivanov**                                              *ivanovserg990@gmail.com*
*Criteo*

**Jun Pang**                                                   *jun.pang@uni.lu*
*University of Luxembourg*

**Reviewed on OpenReview:** *https://openreview.net/forum?id=JBuCfkmKYu*

## Abstract

Graph Neural Networks (GNNs) have been predominant for graph learning tasks; however, recent studies showed that a well-known graph algorithm, Label Propagation (LP), combined with a shallow neural network can achieve comparable performance to GNNs in semi-supervised node classification on graphs with high homophily. In this paper, we show that this approach falls short on graphs with low homophily, where nodes often connect to the nodes of the opposite classes. To overcome this, we carefully design a combination of a base predictor with LP algorithm that enjoys a closed-form solution as well as convergence guarantees. Our algorithm first learns the class compatibility matrix and then aggregates label predictions using LP algorithm weighted by class compatibilities. On a wide variety of benchmarks, we show that our approach achieves the leading performance on graphs with various levels of homophily. Meanwhile, it has orders of magnitude fewer parameters and requires less execution time.

## 1 Introduction

Following the triumph of deep learning in computer vision and natural language processing, more and more success stories are coming from message-passing Graph Neural Networks (GNNs) suited for relational data such as graphs or meshes (Zhang et al., 2020; Wu et al., 2021). The majority of modern deep learning architectures can be considered as a special case of the GNN with specific geometrical structures (Bronstein et al., 2021). These models have achieved state-of-the-art performance in tasks such as (semi-)supervised node classification, common in real-world applications, and crested popular leaderboards such as Open Graph Benchmark (Hu et al., 2020). The landscape of GNNs is rich, and many new architectures have been recently proposed to compensate for limited expressivity (Velickovic et al., 2018; Xu et al., 2019; Du et al., 2019; Azizian & Lelarge, 2020) or to solve specific problems such as over-smoothing, inherent to the traditional message-passing layers (Li et al., 2018; Zhao & Akoglu, 2020; Min et al., 2020; Yan et al., 2021). Unfortunately, these models attain desiderata with the extra price of being more complex and less intuitive during inspection of their performance gains, therefore restricting their applicability in practice.

To address these problems, several models were proposed recently that do not use message-passing algorithm of GNNs but instead are based on well-studied algorithms that show promising results in graph problems (Tian et al., 2019; Rossi et al., 2020; Huang et al., 2021; Ivanov & Prokhorenkova, 2021). Here, we resort to a graph algorithm called Label Propagation (LP) (Zhou et al., 2003; Zhu, 2005) – a competitive algorithm in semi-supervised node classification setup, which was popular for more than a decade. While GNNs learn mapping functions between node features and class labels, LP algorithm directly incorporates

class labels of the train nodes to make predictions on the test nodes. As traditional LP algorithm does not use node features (which may contain significant signal about the class labels of the nodes), it was recently shown (Huang et al., 2021) that by making "base predictions" by a linear network on the node features and then substituting the predictions to the LP algorithm, it is possible to boost the performance up to the results of more complex GNNs. These results, however, are often obtained for the graph datasets that exhibit only high homophily, i.e. structure where neighbouring nodes are likely to have the same class labels. In graphs with low homophily, known as heterophily ("opposites attract"), LP and traditional GNNs fall short and are often outperformed by simple methods such as multi-layer perceptron (Rosenblatt, 1961) (shown in Section 6.3). In order to give a precise description of the node label relationship of an arbitrary graph, here we introduce and formally define the homophily ratio of a graph.

**Definition 1** (Homophily Ratio $h$). *For an arbitrary graph $\mathcal{G} = (\mathcal{V}, \mathcal{E}, \mathbf{X})$, its homophily ratio $h$ is determined by the relationship between node class labels and graph structure encoded in the adjacency matrix. Recent work commonly use two homophily metrics: edge homophily $h_{edge}$ (Zhu et al., 2021) and node homophily $h_{node}$ (Pei et al., 2020). They can be formulated as:*

$$h_{edge} = \frac{|\{(u,v) : (u,v) \in \mathcal{E} \wedge y_u = y_v\}|}{|\mathcal{E}|} \qquad h_{node} = \frac{1}{|\mathcal{V}|} \sum_{v \in \mathcal{V}} \frac{|\{u : u \in \mathcal{N}_v \wedge y_u = y_v\}|}{|\mathcal{N}_v|} \qquad (1)$$

*where $\mathcal{N}_v$ is the set of adjacent nodes of node $v$ and $|\cdot|$ represents the number of elements of the set. Specifically, $h_{edge}$ evaluates the fraction of edges in a graph that connect nodes that have the same class labels; $h_{node}$ evaluates the overall fraction of neighbouring nodes that have the same class labels. In this paper, we focus on edge homophily and set $h = h_{edge}$ in the following sections.*

Motivated by this limitation, several GNN architectures were proposed to make message-passing paradigm work on heterophilous graphs (Zhu et al., 2020; Chen et al., 2020; Yan et al., 2021; Bo et al., 2021; Zheng et al., 2022). These models revolve around modifications of neighbourhoods used for aggregation schemes of GNNs to enrich the diversity of class labels among neighbours. For example, Zhu et al. (2020) uses multiple-hop neighbourhoods for the aggregation in GNNs, which in turn provides more complete information about the connectivity of different classes. While such approaches bridge the gap for traditional GNNs on heterophilous graphs, they often do so at the expense of more parameters and longer training time.

Instead, in this work, we modify LP algorithm to work well in semi-supervised node classification on heterophilous graphs. We start by conducting an experimental investigation over existing models' micro-level performance, i.e., evaluating the node classification accuracy for node groups with subgraphs of different homophily ratios. The investigation results (as shown in Fig. 1) demonstrate that recent GNNs designed for heterophilous graphs do not outperform simple neural network model that only relies raw on node features, i.e., multi-layer perceptron, when the subgraph homophily ratio of a node is low. Inspired by this finding, we propose an efficient framework that relies on base predictions given by a simple neural network and further ameliorate the base predictions with a compatible LP algorithm. In particular, we propose a simple pipeline (CLP) with three main steps (Fig. 2): (i) base predictions of all nodes are made by a simple neural network purely on the node features; (ii) a global compatibility matrix that computes connectivity of different class labels is estimated; and (iii) smoothing of the predictions across neighbours weighted by the compatibility of the class labels is performed. Intuitively, step (i) calculates the class probabilities for the test nodes, while step (ii) defines the weights on edges with which LP algorithm at step (iii) will propagate the class probabilities for each node. While steps (i) and (iii) have been tried independently for semi-supervised classification before (Kipf & Welling, 2017; Ivanov & Prokhorenkova, 2021), it is learning the compatibility matrix at step (ii) that makes a big difference as we show in the experiments. In our theoretical analysis, we show that our approach can be computed via closed-form solution that provides necessary and sufficient conditions for convergence. Empirically, extensive experimental results on a wide variety of benchmarks show the competitive and efficient performance of CLP.

A significant boost in the performance of our method is related to learning a global compatibility matrix between classes. This idea is not new – before the rise of neural networks for semi-supervised learning several algorithms such as DCE (P. et al., 2020), ZooBP (Eswaran et al., 2017), LinBP (Gatterbauer et al., 2015) and FaBP (Koutra et al., 2011) use compatibility matrix for belief-propagation algorithm. However,

all of these methods are motivated by the regularisation framework, where the labelling function minimises some energy objective that does not depend on the node features (Gatterbauer, 2014) and were shown to have suboptimal performance to GNNs (P. et al., 2020). More recently, compatibility matrix was used for GNNs in the heterophily setting (Zhu et al., 2021) and showed a significant increase in performance. That being said, we find that learning a compatibility matrix from the node features significantly improves the performance of LP on heterophilous graphs.

Overall, we generalise LP algorithm to arbitrary heterophily assumption, where the commonly used smoothness assumption (homophily) is a special case with the identity matrix acting as the compatibility matrix. In this case, LP is orders of magnitude faster than log-likelihood estimators such as GNNs, and it presents new ways to understand the performance of graph learning through the lens of diffusion-based learning (Koutra et al., 2011; Gatterbauer, 2014; Zhou et al., 2003; Zhu, 2005). For example, the insights of using compatibility matrix and class labels as part of the training can be incorporated into existing GNN models. As such, we hope that the ideas of LP algorithm could be fruitful for other tasks such as node regression, and LP could become a commonly used baseline of graph learning practitioners.

## 2 Additional Related Work

**GNNs for heterophily regime.** The realisation that standard message-passing Graph Neural Network (GNNs) (Kipf & Welling, 2017; Velickovic et al., 2018; Battaglia et al., 2018) are suboptimal for graphs with high heterophily was not immediate. At first, there was rich literature on solving the over-smoothing problem (Li et al., 2018) which prevents an increasing number of layers of GNNs without loss of performance (common to deep convolutional nets). After that, with new graph datasets with high heterophily (Pei et al., 2020; Abu-El-Haija et al., 2019; Lim et al., 2021; Zheng et al., 2022) and new theory that connects the over-smoothing problem with the tendency of nodes to connect to the opposite classes (Yan et al., 2021), it has become evident that GNNs must incorporate additional knowledge to be suited for heterophilous graphs. Several GNNs were proposed to deal with heterophily setting (Chien et al., 2021; Bo et al., 2021; Zhu et al., 2021); however, usually improved accuracy of these GNNs is traded with an extra computational cost which makes it hard to scale for large datasets, unlike Label Propagation (LP) algorithm, which is a simple graph algorithm. Additionally, Wang & Leskovec (2020) use label propagation as regularisation to assist message-passing GNNs in learning proper edge weights, but their approach is still tailored only for homophilous datasets. A recent approach CPGNN (Zhu et al., 2021) uses compatibility matrix with message-passing process; however, there are several notable differences compared to our approach. First, CPGNN adjusts the weight of the message only based on the class of a sending node and compatibility matrix. In turn, we additionally consider the class of a receiving node, which significantly improves the results in our experiments. Second, we provide additional theoretical analysis of our method, giving a closed-form solution and convergence guarantees, which is not available for CPGNN model.

**Label propagation for heterophily regime.** Perhaps the closest work to ours is (Gatterbauer, 2014; Gatterbauer et al., 2015), where a compatibility matrix is used in the Linearised Belief Propagation (LinBP) algorithm. There, a compatibility matrix is provided or estimated via a closed-form solution to minimise a convex energy function and does not use the node features that are crucial in the estimation of the right labelling functions. Several follow-ups aimed to generalise LinBP to various types of heterophily (Peel, 2017) or Markov Random Fields (Gatterbauer, 2017). It was later shown in the experiments that these methods are less effective than GNNs in graphs with node features (P. et al., 2020). In contrast, our method combines two orthogonal sources of information – one from the labelling function learned on the node features and another from LP algorithm that uses known labels together with the graph structure.

## 3 Preliminaries

An unweighted graph with $n$ nodes can be formally represented as $\mathcal{G} = (\mathcal{V}, \mathcal{E}, \mathbf{X})$, where $\mathcal{V}$ is the set of nodes, $\mathcal{E} \subseteq \mathcal{V} \times \mathcal{V}$ denotes the set of edges, and $\mathbf{X} = \{\mathbf{x}_1, \mathbf{x}_2, \ldots, \mathbf{x}_n\}$. $\mathbf{x}_v \in \mathbb{R}^{\kappa}$ represents node features ($\kappa$ is the dimensionality of node features). $\mathcal{Y}$ stands for the set of possible class labels for $v \in \mathcal{V}$. For subsequent discussion, we summarise $\mathcal{V}$ and $\mathcal{E}$ into adjacency matrix $\mathbf{A} \in \{0,1\}^{n \times n}$.

**Problem setup**. In this paper, we focus on the semi-supervised node classification task on a graph $\mathcal{G}$, where $\mathcal{T}_\mathcal{V} \subset \mathcal{V}$ with known class labels $y_v$ for all $v \in \mathcal{T}_\mathcal{V}$. We aim to infer the unknown class labels $y_u$ for all $u \in \mathcal{V} \setminus \mathcal{T}_\mathcal{V}$. In addition, $\mathcal{T}_\mathcal{V}$ is split into two subsets: $\mathcal{T}_\mathcal{V}^t$ and $\mathcal{T}_\mathcal{V}^v$, where $\mathcal{T}_\mathcal{V}^t$ is training set and $\mathcal{T}_\mathcal{V}^v$ works as the validation set for early stopping or parameter fine-tune to prevent overfitting.

The homophily ratio $h$ defined in Definition 1 is suitable for measuring the *overall* homophily level in the graph. However, the actual homophily level is not necessarily uniform within all parts of the graph. One typical case is that the homophily level varies among different pairs of classes. To measure the variability of the homophily level, we further define the *compatibility matrix* **H** by measuring the fraction of outgoing edges from a node in class $i$ to a node in class $j$.

**Definition 2** (Compatibility Matrix **H**). *The compatibility matrix* **H** *has entries* $[\mathbf{H}]_{ij}$ *that capture the fraction of outgoing edges from a node in class $i$ to a node in class $j$:*

$$[\mathbf{H}]_{ij} = \frac{|(u,v) : (u,v) \in \mathcal{E} \wedge y_u = i \wedge y_v = j|}{|(u,v) : (u,v) \in \mathcal{E} \wedge y_u = i|} \tag{2}$$

The example of Appendix A gives an intuitive explanation of how **H** measures the variability of the homophily level. In the semi-supervised node classification settings, compatibility matrix **H** empirically models the probability of nodes belonging to each pair of classes to connect. Modelling **H** is crucial for heterophily settings, but calculating the exact **H** would require knowledge to class labels of all nodes in the graph, which violates the semi-supervised node classification setting. Therefore it is not possible to incorporate exact **H**. To fill this gap, in Sec. 5.2, we propose an approach to estimate **H** based on a sparsely labelled graph, which is utilised after to assist the label propagation step (Sec. 5.3). An empirical study in Sec. 6.5 empirically discusses the quality of estimated **H** and its influence on the model performances.

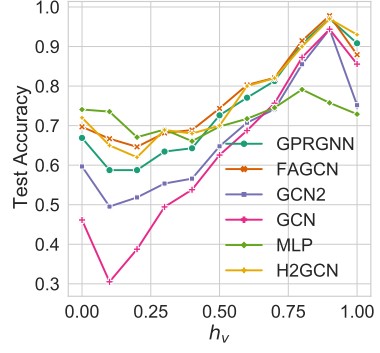

(a) Wiki

## 4 An Experimental Investigation

In this section, we conduct an empirical study to motivate the design of our approach. Unlike the classic *macro*-level node classification evaluation method, we provide a different way to understand existing models' *micro*-level effectiveness. The main idea of this experiment is to study how different models perform at the level of an individual node depending on the homophily ratio of the 1-hop subgraphs. We define homophily ratio of an individual node $h_v$ as follows:

$$h_v = \frac{|\{(u,v) : (u,v) \in \mathcal{E}_v \wedge y_u = y_v\}|}{|\mathcal{E}_v|} \tag{3}$$

where $\mathcal{E}_v$ is the edge set of the induced 1-hop neighbourhood of $v$.

We take two graphs as examples: a heterophily graph Wiki with $h = 0.30$ and a homophily graph ACM with $h = 0.82$. Following the *medium splitting* (Sec. 6.2 includes details settings), we train different models on the training nodes of a graph and compute predictions for the test nodes. We then aggregate the accuracy of predictions for each level of homophily,

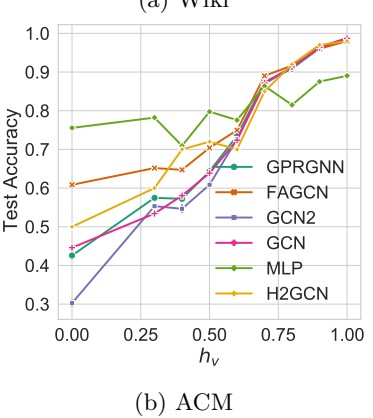

(b) ACM

Figure 1: Classification accuracy for different 1-hop subgraph homophily ratios on Wiki (1a) and ACM (1b) graphs.

$\{0, 0.1, \ldots, 0.9, 1\}$, and plot the obtained results in Fig. 1. Global accuracy across all test nodes can be found in Tab. 2 and Fig. 4.

Results from Tab. 2 and Fig. 4 demonstrate that in general GNNs outperform multi-layer perceptron (MLP) (Rosenblatt, 1961). However, if we zoom in on local neighbourhoods, as shown in Fig. 1, the results of MLP are often better than those of GNNs when the homophily ratio of a node's 1-hop subgraph is low.

In particular, we can see from Fig. 1 that (i) vanilla GCN has superior accuracy for nodes with strong subgraph homophily ratio ($h_v \geq 0.7$) on both graphs; other advanced GNN models mainly improve the classification accuracy over nodes with low $h_v$, and (ii) MLP is relatively stable across different homophily ratio $h_v$ and is a better model for nodes with low $h_v$ compared to other GNNs. For instance, MLP achieves the best accuracy on nodes with $h_v \leq 0.3$ on Wiki graph and nodes with $h_v \leq 0.6$ on ACM graph.

This illustrates that recent GNN models designed for heterophilous graphs do not outperform MLP for nodes with a considerable fraction of neighbours with opposite class labels; instead, they have better global accuracy than MLP by having better accuracy on nodes with high homophily ratio $h_v$. Based on this evidence, we propose a simple but effective approach that mainly relies on the predictions of MLP to maintain its favourable performance on nodes with low homophily $h_v$ and that further ameliorates the classification results by incorporating the knowledge of the graph structure.

## 5 Compatible Label Propagation with Heterophily

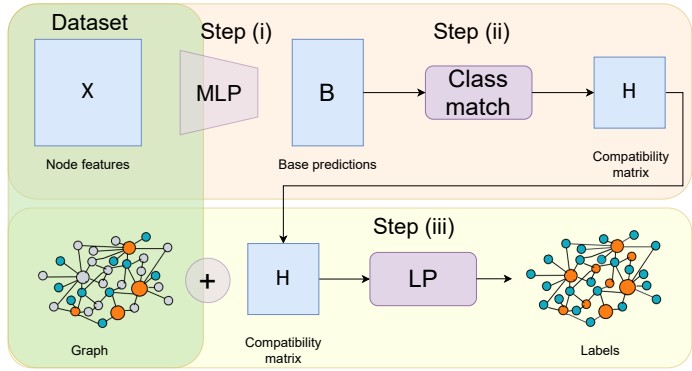

Figure 2: Overview of Compatible Label Propagation (CLP) model. Step (i): base predictor, MLP, makes class predictions for each node using only node features. Step (ii): global compatibility matrix between classes is computed with Eq. 6. Step (iii): propagate class predictions with LP algorithm and get the classes for test nodes. Intuitively, compatibility matrix measures the weighted probabilities of any two target classes being connected, and as such, it defines the edge weights in LP algorithm.

Our approach starts with a simple base predictor on raw node features, which does not rely on any learning over the topological structure. Any off-the-shelf graph-agnostic model can be plugged in to become a base predictor, which enables our approach to accommodate any node features. After, we propose an approach to estimate the compatibility matrix $\mathbf{H}$ of the overall graph and apply it to calculate the relation between each pair of nodes. Finally, we use label propagation algorithm with an estimated compatibility matrix to smooth the prior prediction probabilities on the weighted graph to get the final predictions.

### 5.1 Simple Base Predictor

To start, we use a simple base predictor that does not rely on graph structure to learn prior predictions. Specifically, we train a model $f_\theta$ to minimise $\sum_{v \in \mathcal{T}_\mathcal{V}^t} \mathcal{L}(f_\theta(\mathbf{x}_v), y_v)$, where $\mathbf{x}_v$ is the available feature of node $v$ and $y_v$ is its true class label, $\mathcal{L}$ is a loss function. In this paper, we adopt a simple multi-layer perceptron (MLP) (Rosenblatt, 1961) as the base predictor, where $\ell$-th layer can be formally formulated as following:

$$\mathbf{D}^{(\ell)} = \sigma(\mathbf{D}^{(\ell-1)}\mathbf{W}^{(\ell)} + \mathbf{b}^{(\ell)}) \tag{4}$$

where $\mathbf{W}^{(\ell)}$ are learnable parameters and $\mathbf{b}^{(\ell)}$ is the bias vector. $\sigma$ is the activation function (e.g. ReLU), and we initialise $\mathbf{D}^{(0)} = \mathbf{X}$.

From $f_\theta$, we get a prior prediction $\hat{\mathbf{D}} = \text{Softmax}(\mathbf{D}^{(L)}) \in \mathbb{R}^{|\mathcal{V}| \times |\mathcal{Y}|}$, where $\ell = L$ is the last layer. Omitting the graph for the prior predictions brings several benefits: (i) it avoids the sensitivity to homophily/heteriophily

of the graph (as was shown in Fig. 5, MLP's performance maintains good stability for graphs with different homophily ratios); and (ii) it significantly reduces the number of parameters that we need to learn, thus accelerating the approach (as shown in Fig. 8). Next, we use MLP's predictions to estimate the weights for label propagation algorithm.

## 5.2 Estimation of Compatibility Matrix

The focal idea of compatibility matrix is summarising the relative frequencies of classes between neighbours. Under the semi-supervised node classification settings, we only know the class labels of a small fraction of nodes $(\mathcal{T}_\mathcal{V}^t)$. We derive the preliminary class labels of unknown nodes $(\mathcal{V} \setminus (\mathcal{T}_\mathcal{V} \cup \mathcal{T}_\mathcal{V}^v))$ as the base prediction $\hat{\mathbf{D}}$. Note that we treat validation set nodes as unknown nodes, which will be used to evaluate the performance of LP step and pick up the better final predictions. More specifically, denote the training mask $\mathbf{M}$ as: $[\mathbf{M}]_{i,:} = \begin{cases} \mathbf{1}, & \text{if } i \in \mathcal{T}_\mathcal{V}^t \\ \mathbf{0}, & \text{otherwise} \end{cases}$ . The preliminary knowledge of class labels can be formally represented as:

$$\hat{\mathbf{B}}^{(0)} = \mathbf{M} \circ \mathbf{Y} + (1 - \mathbf{M}) \circ \hat{\mathbf{D}} \tag{5}$$

where $\circ$ is the *Hadamard* (element-wise) product, $\mathbf{Y} \in \mathbb{R}^{|\mathcal{V}| \times |\mathcal{Y}|}$ and $\mathbf{Y}_{vj} = 1$ if $y_v = j$, otherwise $\mathbf{Y}_{vj} = 0$.

Next, we estimate a compatibility matrix $\hat{\mathbf{H}}$ that calculates the probability that a training node of one class is connected with a node of another class.

$$\hat{\mathbf{H}} = \mathcal{S}((\mathbf{M} \circ \mathbf{Y})^\top \mathbf{A} \hat{\mathbf{B}}^{(0)}) \tag{6}$$

where $\mathcal{S}$ is the Sinkhorn-Knopp function that ensures $\hat{\mathbf{H}}$ is doubly-stochastic (Sinkhorn & Knopp, 1967).

A compatibility matrix $\hat{\mathbf{H}}$ can be seen as a multiplication of two matrices, $(\mathbf{M} \circ \mathbf{Y})^\top$ and $\mathbf{A} \hat{\mathbf{B}}^{(0)}$. The matrix $(\mathbf{M} \circ \mathbf{Y})^\top$ represents one-hot encoded class labels of training nodes only. In turn, the matrix $\mathbf{A} \hat{\mathbf{B}}^{(0)}$ computes the sum of class probabilities across all neighbours of each node. After multiplication of these two matrices, each entry $(i, j)$ of $(\mathbf{M} \circ \mathbf{Y})^\top \mathbf{A} \hat{\mathbf{B}}^{(0)}$ represents a score that a class $i$ among training nodes is connected with a node of class $j$ estimated with prior probabilities $\hat{\mathbf{D}}$. A function $\mathcal{S}$ converts these scores back to probabilities such that each entry $(i, j)$ of $\hat{\mathbf{H}}$ indicates a probability that a class $i$ is connected with class $j$.

## 5.3 Compatible Label Propagation

After obtaining the estimation $\hat{\mathbf{H}}$, we propagate the knowledge about node class labels with the guide of $\hat{\mathbf{H}}$ over the graph. The key idea of our method is that the edge weight of a message $u \mapsto v$ in label propagation algorithm depends on both predicted classes of sending and receiving nodes. That contrasts with previous works (Gatterbauer et al., 2015; Zhu et al., 2021) where edge weight depends only on the sending node class probabilities. In particular, for each edge $(i, j)$, we define an edge weight as follows:

$$[\mathbf{F}]_{ij} = ([\hat{\mathbf{B}}^{(0)}]_i \hat{\mathbf{H}}) \circ [\hat{\mathbf{B}}^{(0)}]_j \tag{7}$$

Intuitively, edge weight $[\mathbf{F}]_{ij}$ depends on the probabilities that node $i$ is connected with some class $k$, $([\hat{\mathbf{B}}^{(0)}]_i \hat{\mathbf{H}})$, and the probabilities that node $j$ has the same class $k$. Naturally, we can assign the edge weights to corresponding positions of adjacent matrix to get $\mathbf{A}^{\mathbf{F}} \in \mathbb{R}^{n \times n \times |\mathcal{Y}|}$, where $[\mathbf{A}]_{ij}^{\mathbf{F}} = [\mathbf{F}]_{ij}$.

Let $\hat{\mathbf{B}}$ and $\hat{\mathbf{D}}$ be the final node class prediction matrix and the base prediction, respectively. $\mathbf{A}^{\mathbf{F}}$ is the fixed weighted adjacent matrix. Then, the final node classifications are approximated by the equation system:

$$\hat{\mathbf{B}} = (1 - \alpha)\hat{\mathbf{D}} + \alpha \, \mathbf{A}^{\mathbf{F}} \oplus \hat{\mathbf{B}} \tag{8}$$

where $[\mathbf{A}^{\mathbf{F}} \oplus \hat{\mathbf{B}}]_{:,k} = \mathbf{A}_k^{\mathbf{F}} [\hat{\mathbf{B}}]_{:,k}$ and $\mathbf{A}_k^{\mathbf{F}}$ means the weighted adjacent matrix with $k$-th dimensional edge weights. $\alpha$ is a hyperparameter, which defines how much update to the previous state each label propagation step makes.

**Iterative updates.** Notice that Eq. 8 gives an implicit definition of the final node classification after convergence, it can also be used as iterative update equations, allowing an iterative calculation of the final node classification predictions:

$$\hat{\mathbf{B}}^{(r+1)} \leftarrow (1 - \alpha)\hat{\mathbf{D}} + \alpha\mathbf{A}^{\mathbf{F}} \oplus \hat{\mathbf{B}}^{(r)} \tag{9}$$

Thus, the final node classification predictions can be computed via linear matrix operations. Note that previous works (Gatterbauer et al., 2015; Zhu et al., 2021) compute the compatibility matrix $\hat{\mathbf{H}}$ for LP as follows:

$$\hat{\mathbf{B}}^{(r+1)} \leftarrow (1 - \alpha)\hat{\mathbf{D}} + \alpha\mathbf{A}\hat{\mathbf{B}}^{(r)}\hat{\mathbf{H}} \tag{10}$$

Eq. 10 defines an edge weight by the relation between the sending node and $\hat{\mathbf{H}}$. Hence, receiving nodes get the same message from a sending node regardless of the class of the receiving nodes. We argue that the proper weight of a message should be determined by both sending and receiving nodes (Fig. 3). Appendix A presents a detailed comparison between different LP-related methods, and we empirically demonstrate the advantages of CLP in Sec. 6.5.

### 5.4 Theoretical Analysis of CLP

Eq. 9 allows solving CLP Eq. 8 via iterative updates. Here, we show an alternative method that provides a closed-form solution, which in turn sets convergence guarantees of CLP for each class $k$. We start by defining vectorisation of a matrix $\mathbf{X}$, which stacks columns of $\mathbf{X}$ side-by-side.

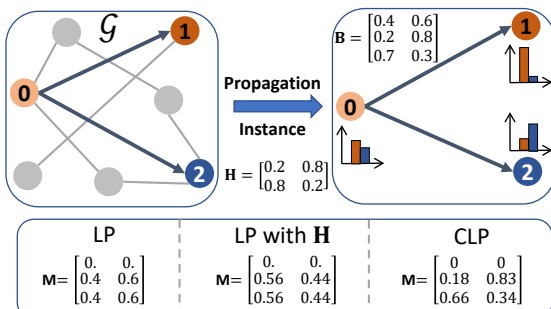

Figure 3: Comparison of three propagation schemes, $\mathbf{M}$ represents the received messages after one propagation iteration. In LP nodes **1** and **2** receive the same message; LP with $\mathbf{H}$ overturns the prior prediction of node **1**; CLP adapts the heterophily of the graph and reassures confident prior predictions.

**Definition 3** (Matrix Vectorisation (H. V. Henderson, 1981)). *Vectorisation of an $m \times n$ matrix $\mathbf{X}$ is an $mn \times 1$ vector given by:*

$$vec(\mathbf{X}) = [\mathbf{x}_{11}, \ldots, \mathbf{x}_{n1}, \mathbf{x}_{12}, \ldots, \mathbf{x}_{n2}, \ldots, \mathbf{x}_{1n}, \ldots, \mathbf{x}_{nn}]^{\mathrm{T}} \tag{11}$$

Additionally, the Kronecker product of $\mathbf{X}$ and $\mathbf{Q}$ is the $mp \times nq$ matrix is defined by:

$$\mathbf{X} \otimes \mathbf{Q} = \begin{bmatrix} \mathbf{x}_{11}\mathbf{Q} & \mathbf{x}_{12}\mathbf{Q} & \ldots & \mathbf{x}_{1n}\mathbf{Q} \\ \mathbf{x}_{21}\mathbf{Q} & \mathbf{x}_{22}\mathbf{Q} & \ldots & \mathbf{x}_{2n}\mathbf{Q} \\ \vdots & \vdots & \ddots & \vdots \\ \mathbf{x}_{m1}\mathbf{Q} & \mathbf{x}_{m2}\mathbf{Q} & \ldots & \mathbf{x}_{mn}\mathbf{Q} \end{bmatrix} \tag{12}$$

We are now ready to give a closed-form solution to Eq. 8:

**Proposition 1** (Closed-form CLP). *The closed-form solution for CLP (Eq. 8) for class $k$ is given by:*

$$vec([\hat{\mathbf{B}}]_{:,k}) = (\mathbf{I} - \alpha\,(\mathbf{I} \otimes \mathbf{A}_k^{\mathbf{F}}))^{-1}(1 - \alpha)\,vec([\hat{\mathbf{D}}]_{:,k}) \tag{13}$$

Proof of Proposition 1 refers to Appendix B.

Therefore, instead of iterative updates Eq. 10, we can compute the final node predictions in a closed-form by using Eq. 13, as long as the inverse of the matrix exists. Based on this closed-form solution we next establish necessary and sufficient criteria for convergence.

**Convergence of iterative CLP**. We remind that spectral radius of a matrix $\mathbf{X}$ is the maximum eigenvalue, i.e. $\rho(\mathbf{X}) = \max(\{|\lambda_1|, \ldots, |\lambda_n|\})$. With Eq. 13 we are now ready to establish convergence guarantees for CLP.

**Proposition 2** (Convergence of CLP). *For class $k$, CLP iterative updates Eq. 13 converge if and only if $\rho(\mathbf{A}_k^{\mathbf{F}}) < \alpha^{-1}$.*

Proof of Proposition 2 refers to Appendix B.

As computing the largest eigenvalue may be too expensive for large graphs, following the Gershgorin circle theorem (Weisstein, 2003), one can replace the spectral norm with any sub-multiplicative norm that is faster to compute and give an upper bound to the spectral radius. For $\| \mathbf{X} \|_p = (\sum_i \sum_j |\mathbf{X}(i,j)|^p)^{1/p}$ we have $\rho(\mathbf{X}) \leq \| \mathbf{X} \|_2 \leq \| \mathbf{X} \|_1$. Hence, one can use Frobenius or 1-induced norm to efficiently check if the sufficient condition for convergence is satisfied. In our experiments, we found that CLP converges for all datasets.

## 5.5 Summary

To review our approach, we start with a base predictor, which purely learns from node features to make node class label predictions. Next, we estimate the global compatibility matrix $\hat{\mathbf{H}}$ based on the sparsely labelled graph and base predictions. $\hat{\mathbf{H}}$ describes the overall possibility of nodes belonging to each pair of classes to connect, which can be utilised to estimate the relationship between each pair of base prediction vectors. Finally, we perform an efficient LP step to smooth base predictions and obtain class labels with the assistance of the relationship between each pair of nodes.

Compared with existing GNN models, CLP similarly benefits from both node features and graph structure, yet separates them into two processes. It is motivated by the investigation in Sec. 4 that MLP has better accuracy over other GNN models for nodes with low homophily $h_v$. Hence we would like to maintain MLP's advantages and utilise graph structure to improve it to obtain final predictions. Following this way, both node features and graph structure are appropriately involved in our approach, and it only requires learning parameters specified by a base predictor. Next, we are going to demonstrate the competitive performances of CLP on node classification tasks.

# 6 Experiments

Table 1: Statistics for six synthetic datasets. (Prod) means contextual node features come from Ogbn-Products, or adopt the statistic features designed by 2D Gaussians.

| Benchmark Name | #Nodes $|\mathcal{V}|$ | #Edges $|\mathcal{E}|$ | #Classes $|\mathcal{Y}|$ | Homophily $h$ | #Avg. Degree |
|---|---|---|---|---|---|
| Syn-(Prod)-1 | $10,000$ | $49,446$ to $50,352$ | 10 | $[0, 0.1, \ldots, 1]$ | 4.95 to 5.02 |
| Syn-(Prod)-2 | $10,000$ | $99,556$ to $99,556$ | 10 | $[0, 0.1, \ldots, 1]$ | 9.96 to 10.01 |
| Syn-(Prod)-3 | $10,000$ | $149,090$ to $15,1494$ | 10 | $[0, 0.1, \ldots, 1]$ | 14.91 to 15.15 |

To validate our approach's effectiveness, we first empirically demonstrate the performance of CLP and state of the art (SOTA) models on real-world and synthetic datasets with a wide variety of settings. Second, we compare the number of required parameters, the quality of compatibility estimation, the models' execution time, and their performance on different graphs with different label rates. Third, we empirically show the advantages of our propagation method compared with the previous design. We also study the influence of different label rates on the compatibility matrix estimation and classification accuracy and show the efficiency of CLP in terms of the model size.

## 6.1 Datasets

**Real-world datasets**. We use a total of 19 real-world datasets (Texas, Wisconsin, Actor, Squirrel, Chameleon, USA-Airports, Brazil-Airports, Wiki, Cornell, Europe-Airports, deezer-europe, Twitch-EN, Twitch-RU, Ogbn-Proteins, WikiCS, DBLP, CS, ACM, Physics) in diverse domains (web-page, citation, co-author, flight transport, biomedical and online user relation). Note that we use ROC-AUC as the evaluation metric for the class imbalanced datasets, i.e., Twitch-EN, Twitch-RU and Ogbn-Proteins, following Lim et al. (2021). For other datasets, we use node classification accuracy as our general evaluation metric. See Appendix C for detailed descriptions, statistics and references.

**Synthetic datasets**. We generate random synthetic graphs with various homophily ratios $h$ and node features by adopting a similar approach (Abu-El-Haija et al., 2019; Kim & Oh, 2021) but with some mod-

ifications. For instance, synthetic graphs (Abu-El-Haija et al., 2019) have no available contextual node attributes. Specifically, each synthetic graph has 10 classes and 1,000 nodes per class. Nodes are assigned random features sampled from 2D Gaussians (Syn) or contextual features from real-world datasets (Hu et al., 2020) (Syn-Prod). Except for the homophily ratio, we also control the average degree of each graph (around 5, 10 or 15) to investigate the performance with respect to graph sparsity. Here, we give detailed descriptions of the generation process.

*Graph generation.* We generate synthetic graph $\mathcal{G}$ of $|\mathcal{V}|$ nodes with $|\mathcal{Y}|$ different class labels, and $\mathcal{G}$ has $|\mathcal{V}|/|\mathcal{Y}|$ nodes per class. $|\mathcal{V}|$ and $|\mathcal{Y}|$ are two prescribed numbers to determine the size of $\mathcal{G}$. A synthetic graph's homophily ratio $h$ is mainly controlled by $p_{in}$ and $p_{out}$, where $p_{in}$ means the possibility of existing an edge between two nodes with the same label and $p_{out}$ is the possibility of existing an edge between two nodes with different class labels. Furthermore, the average degree of $\mathcal{G}$ is $d_{avg} = |\mathcal{V}|/|\mathcal{Y}| \cdot \delta$, where $\delta = p_{in} + (|\mathcal{Y}| - 1) \cdot p_{out}$. Following the described graph generation process, with given $|\mathcal{V}|$, $\mathcal{Y}$ and $d_{avg}$, we choose $p_{in}$ from $\{0.0001\delta, 0.1\delta, 0.2\delta, \ldots, 0.9\delta, 0.9999\delta\}$. Note that the synthetic graph generation process requires both $p_{in}$ and $p_{out}$ are positive numbers, hence we use $p_{in} = 0.0001\delta$ and $0.9999\delta$ to estimate $h = 0$ and $h = 1$ cases, respectively.

*Node features generation.* In order to comprehensively evaluate the performances of different models, we assign each node with statistic features (Syn) or real-world contextual node features (Syn-Prod). For graphs with statistic node features, the feature values of nodes are sampled from 2D Gaussian (Abu-El-Haija et al., 2019). The mean of Gaussian can be described in polar coordinates: each means has radius 300 and angle $\frac{2\pi}{10} \times (class\ id)$. The covariance matrix of each class is $3500 \times diag[7, 2]$, that is rotated by angle $\frac{2\pi}{10} \times (class\ id)$. For datasets with real-world contextual node features, we first establish a class mapping $\psi : \mathcal{Y} \rightarrow \mathcal{Y}_b$ between classes in the synthetic graph $\mathcal{Y}$ to classes of existing benchmark graph $\mathcal{Y}_b$. The only requirement for the target graph dataset is that the class size and node set size in the benchmark is larger than that of the synthetic graph, i.e., $|\mathcal{Y}|_b \leq |\mathcal{Y}|$ and $|\mathcal{V}| \leq |\mathcal{V}|_b$. In this paper, we adopt the large-scale benchmark, Ogbn-Products (Hu et al., 2020).

## 6.2 Experimental Setup

**Baseline methods.** We compare our model against state-of-the-art graph neural networks and related node classification methods for all datasets under fair settings. Specifically, MLP (Rosenblatt, 1961) is the baseline model that only utilises node attributes, while LINK (Zheleva & Getoor, 2009) only utilises graph structure. Meanwhile, we also adopt general GNN models with underlying homophily assumption: GCN (Kipf & Welling, 2017), GAT (Velickovic et al., 2018) and GCN2 (Chen et al., 2020). Moreover, we adopt several models that are designed for heterophily graphs: Mixhop (Abu-El-Haija et al., 2019), SuperGAT (Kim & Oh, 2021), GPRGNN (Chien et al., 2021), FAGCN (Bo et al., 2021), H2GCN (Zhu et al., 2020) and CPGNN (Zhu et al., 2021). At last, two LP-based models: LP (Zhu, 2005) and C&S (Huang et al., 2021).

**Implementation and splits.** We follow the experimental setup of FAGCN and CPGNN with minor adjustments. Specifically, our experimental setup examines the semi-supervised node classification in the transductive setting. We consider three different choices for the random split into training/validation/test settings, which we call *sparse* splittings (5%/5%/90%), *medium* splitting (10%/10%/80%) and *dense* splitting (48%/32%/20%), respectively. The *sparse* splitting (5%/5%/90%) is similar to the original semi-supervised setting in Kipf & Welling (2017), but we do not restrict each class to have the same number of training instances since it is the case closer to the real-world application. For a fair comparison, we generate 10 fixed split instances with different splitting and results are summarised after 10 runs with random seeds. Note that the Ogbn-Proteins dataset adopts its default splitting settings. Other model setups and hyperparameter settings can be found in Appendix E. Our implementation is available at `https://github.com/zhiqiangzhongddu/TMLR-CLP`.

## 6.3 Results on Real-world Graphs

**Real-world graphs with *heterophily*.** The performance of diverse methods on heterophily graphs under *medium* splitting is summarised in Tab. 2, top-2 performances of each graph are highlighted in colour. Advanced GNN models that are designed for heterophily graphs generally perform better than GNNs designed

Table 2: Summary of node classification results on *heterophily* graphs under *medium* splitting. ‡ indicates the results from Lim et al. (2021). Top-2 performances per benchmark are highlighted in ▮ and ▮, respectively.

| Hom.R $h$ | Texas 0.06 | Wisconsin 0.17 | Actor 0.22 | Squirrel 0.22 | Chameleon 0.23 | USA-A. 0.25 | Bra.-A. 0.29 | Wiki 0.30 | Cornell 0.30 | Eu.-A. 0.31 | deezer 0.53 | Tw.-EN 0.60 | Tw.-RU 0.639 | O.-Proteins − | Rank |
|---|---|---|---|---|---|---|---|---|---|---|---|---|---|---|---|
| MLP | 67.94±3.87 | 69.32±3.33 | 32.07±0.72 | 26.18±0.81 | 35.94±1.47 | 54.92±2.34 | 59.52±11.66 | 70.13±1.18 | 68.19±1.55 | 50.41±3.24 | 63.77±0.30 | 59.56±0.92 | 49.33±1.55 | 73.43±0.12‡ | 3 |
| LINK | 59.52±4.11 | 47.79±7.05 | 24.03±0.61 | 46.02±0.96 | 58.28±1.53 | 24.71±1.17 | 27.97±4.42 | 25.07±1.16 | 46.47±12.28 | 29.59±3.55 | 55.95±0.34 | 55.65±1.02 | 51.27±1.20 | 63.49±0.02‡ | 10 |
| GCN | 54.17±3.18 | 47.55±3.50 | 26.82±0.97 | 24.71±0.86 | 34.61±2.93 | 30.88±2.42 | 26.84±5.84 | 53.15±1.51 | 55.81±1.54 | 31.65±4.61 | 59.94±0.55 | 59.79±0.55 | 51.51±1.05 | 72.03±0.32‡ | 10 |
| GAT | 54.12±3.25 | 48.73±3.32 | 27.37±1.03 | 24.55±0.90 | 36.60±2.30 | 28.13±4.19 | 23.76±1.57 | 47.21±1.60 | 55.18±2.54 | 24.34±1.21 | 56.22±1.17 | 58.66±0.91 | 51.65±1.78 | OOM‡ | 12 |
| GCN2 | 55.22±6.17 | 47.63±4.61 | 27.14±0.65 | 25.5±2.08 | 36.26±2.72 | 36.59±3.01 | 27.22±5.35 | 60.29±3.17 | 53.87±6.38 | 35.05±5.86 | 62.33±0.81 | 59.66±0.45 | 51.53±2.36 | 74.10±0.59 | 8 |
| Mixhop | 54.62±3.49 | 51.63±4.36 | 27.46±1.39 | 27.81±1.13 | 38.14±2.10 | 52.68±1.56 | 44.41±8.22 | 61.74±2.20 | 51.29±7.12 | 45.55±3.88 | 64.16±0.85 | 60.38±0.99 | 52.54±1.49 | 75.60±0.85‡ | 5 |
| SuperGAT | 54.88±2.84 | 49.94±3.20 | 26.69±0.62 | 24.88±1.05 | 35.49±2.26 | 27.02±3.86 | 23.47±1.97 | 33.23±1.79 | 54.47±1.79 | 24.63±1.21 | 57.07±0.64 | 59.66±0.46 | 50.95±1.88 | OOM | 13 |
| GPRGNN | 55.31±3.29 | 50.89±4.00 | 27.72±0.92 | 25.29±1.15 | 34.67±2.82 | 41.83±6.48 | 24.85±3.43 | 68.02±1.30 | 55.03±4.15 | 31.47±5.43 | 62.74±0.39 | 59.42±0.71 | 51.17±1.50 | OOM‡ | 9 |
| FAGCN | 60.95±4.05 | 63.08±5.42 | 32.60±0.85 | 24.93±1.10 | 36.68±1.80 | 56.14±1.34 | 48.19±12.13 | 72.12±0.75 | 62.32±4.32 | 48.22±3.30 | 65.04±0.45 | 60.76±0.74 | 50.19±2.02 | OOM | 2 |
| H2GCN | 61.29±5.20 | 65.67±8.51 | 32.27±0.91 | 26.95±1.74 | 36.93±1.73 | 54.24±1.56 | 38.95±8.06 | 70.57±1.23 | 57.26±6.46 | 40.56±4.78 | 62.82±0.68 | 59.06±0.92 | 51.22±1.33 | OOM‡ | 7 |
| CPGNN | 62.95±15.24 | 70.05±7.30 | 32.42±0.65 | 28.70±1.41 | 47.7±2.04 | 25.21±1.01 | 27.51±6.56 | 70.18±1.13 | 68.04±5.85 | 34.86±1.90 | 64.95±0.39 | 57.07±1.28 | 52.37±0.34 | OOM | 4 |
| LP | 15.58±5.47 | 11.40±3.25 | 17.69±0.57 | 17.59±1.30 | 20.62±2.02 | 24.35±1.31 | 24.48±3.22 | 23.89±0.77 | 18.51±3.19 | 27.20±1.74 | 55.44±0.46 | 54.42±0.81 | 51.90±1.40 | 75.14±0.00‡ | 14 |
| C&S | 66.90±6.60 | 67.34±7.47 | 31.94±1.30 | 26.85±0.94 | 26.85±0.94 | 45.26±4.70 | 55.33±9.31 | 71.49±1.27 | 67.04±5.29 | 37.32±5.50 | 63.92±0.71 | 59.36±1.84 | 52.12±0.83 | 71.13±0.69‡ | 5 |
| CLP (Ours) | 69.63±3.75 | 72.64±5.79 | 33.1±0.65 | 31.76±1.03 | 43.29±1.10 | 56.3±1.44 | 63.53±11.00 | 74.08±2.03 | 70.36±4.83 | 53.83±2.63 | 65.69±0.32 | 60.81±0.78 | 52.78±0.79 | 75.73±0.24 | 1 |

with high-homophily assumption. MLP, which only utilises node features, achieves outstanding performances in several benchmarks. Our model, CLP, inherits the advantage of MLP but also benefits from graph structure, and it achieves outstanding and stable performance on all heterophily graphs. Moreover, many baseline methods lead to out-of-memory (OOM) issues on the large dataset, i.e., Ogbn-Proteins, but CLP avoids this problem, demonstrating its memory efficiency.

**Real-world graphs with *homophily*.** The performance of representative models on homophily graphs under *medium* splitting is summarised in Fig. 4. Inspired by Huang et al. (2021), we further adopt the spectral and diffusion features as additional node features to C&S and CLP and compare their performances with the best performance of SOTA GNN models. C&S$^\dagger$ and CLP$^\dagger$ refer to performance with additional node features and results from the figure demonstrates that CLP$^\dagger$ outperforms or matches the SOTA on *homophily* graphs.

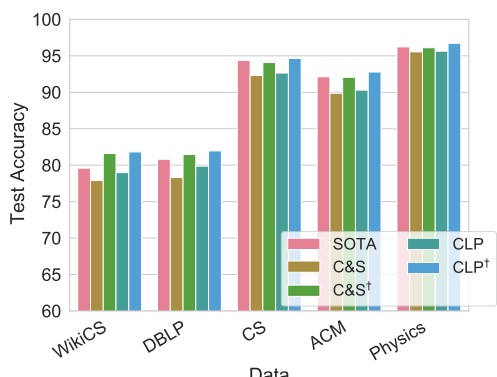

Figure 4: Performance comparison of C&S and CLP with the best performance of GNN models (SOTA) on *homophily* graphs under *medium* splitting.

### 6.4 Results on Synthetic Graphs

**Synthetic graphs *without* contextual node features.** Most previous work (Kipf & Welling, 2017; Bo et al., 2021; Zhu et al., 2020) on semi-supervised node classification has focused only on graphs with contextual features on the nodes. However, the vast majority of graph data does not have node-level contextual features in practical applications, which significantly limits the utility of methods proposed in prior work. Besides, several components of our approach depend on node features. For instance, the compatibility matrix estimation ($\hat{\mathbf{H}}$) relies on the prior predictions which are learned from node features. $\hat{\mathbf{H}}$ plays a crucial role in the following LP step. Therefore, it is natural to ask how CLP performs over graphs without contextual node features compared with other competitive models?

To answer this question, we conduct extensive experiments on semi-supervised node classification with *sparse*, *medium* and *dense* splittings on three synthetic datasets with different average degrees. For instance, the Syn-1 dataset contains 11 graphs with $h$ from 0 to 1, and the average degree per graph is set to around 5 (4.95 to 5.02). Syn-2 and Syn-3 follow similar settings, but the average degree of each graph is set to 10 and 15, respectively.

We present the results of representative models of three synthetic datasets in Fig. 5-(a, b, c). We observe similar trends in three figures: CLP has the best trend overall, outperforming SOTA methods in heterophily settings while matching with other SOTA methods in homophily settings. The performance of vanilla GCN and GCN2 increases with respect to the homophily level ($h \to 1$). But, while synthetic graphs have no contextual node features, MLP is more accurate than them under strong heterophily ($h \to 0$). From Fig. 5, we can find that the classification accuracy of MLP has been stable at about 45%, a relatively low level.

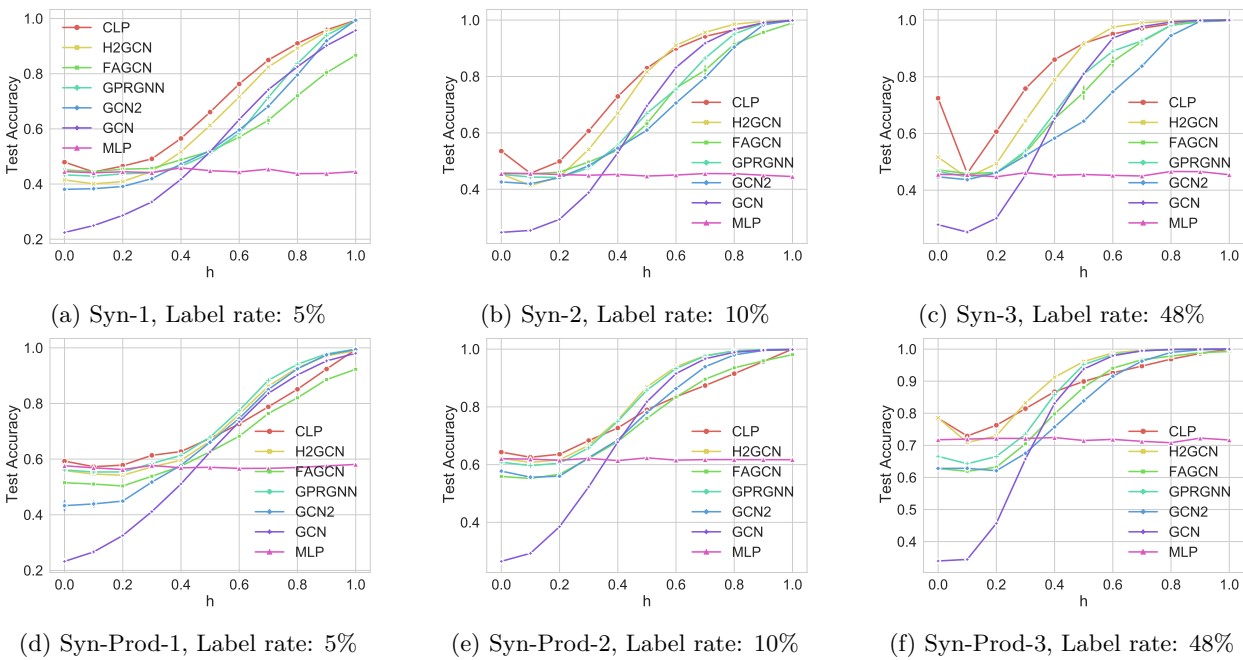

(a) Syn-1, Label rate: 5%     (b) Syn-2, Label rate: 10%     (c) Syn-3, Label rate: 48%

(d) Syn-Prod-1, Label rate: 5%     (e) Syn-Prod-2, Label rate: 10%     (f) Syn-Prod-3, Label rate: 48%

Figure 5: Classification accuracy of different methods with different label rates on synthetic datasets. Only competitive results are presented due to the space limit.

Yet, CLP can still achieve the overall best performance. Overall, it indicates that our approach works for graphs without contextual features.

**Synthetic graphs *with* contextual node features.** We perform extensive experiments on graphs with contextual features to further validate the performance of CLP under various settings.

Similar to the experiments on synthetic graphs *without* contextual node features, there are three synthetic graphs, i.e., Syn-Prod-1, Syn-Prod-2 and Syn-Prod-3, which have the same graph structure as Syn-1, Syn-2 and Syn-3, but *with* contextual node features. Experimental results are presented in Fig. 5-(d, e, f). These figures emphasise that CLP is the best model for most heterophily cases ($h \to 1$), which again confirms the effectiveness of our approach. It echoes the results of the real-world graphs (Tab. 2). Besides, GCN and GCN2, which were proposed with implicit homophily assumption, are significantly less accurate than MLP (near-flat performance curve as it is graph-agnostic) under strong heterophily ($h \le$ 0.4). Such evidence can be found in some cases for other heterophilous GNN models (H2GCN, FAGCN, GPRGNN). For instance, they perform significantly better than GCN but are outperformed by MLP on Syn-Prod-1 under $h \le 0.3$ (Fig. 5d). It reaffirms what we found in Sec. 4, i.e.

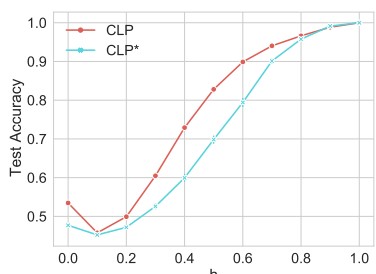

Figure 6: Performance comparison of CLP and CLP* on Syn-1 dataset with *medium* splitting.

MLP could be a better choice for making classification for strong heterophily node groups. Our approach, CLP, can consistently achieve better performance than MLP in graphs with any heterophily levels and sparsity levels.

## 6.5   Additional Analysis

**Comparison between two propagation schemes.** In Sec. 5.3, we explained the design of our compatible LP process and discussed its advantages over prior work (Zhu et al., 2020). The messages between two nodes are adaptively determined by nodes of both ends. Here, we perform extensive experiments to empirically compare the performance of LP steps with two propagation schemes. We choose one synthetic dataset with 11 graphs under various homophily (Syn-1) under the *medium* splitting. Other settings follow the common

setup of CLP as described in Sec. 6.2. The approach that utilises Eq. 10 named CLP[*]. Their performances are reported in Fig. 6. We observe that CLP has the better trend overall, outperforming CLP[*] in most heterophily settings ($h \leq 0.9$) and matching with CLP[*] in other settings.

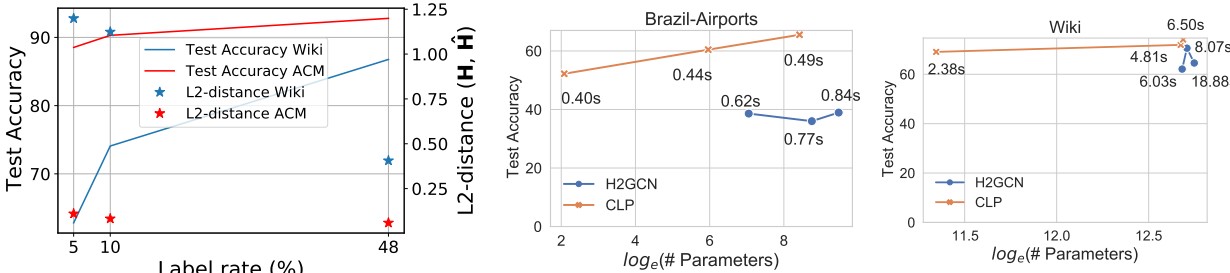

Figure 7: Classification accuracy and L2-distance between estimated/true compatibility matrix with different label rates.

Figure 8: Classification accuracy and execution time of different methods with different layers on *heterophily* graphs. Execution time is marked in the plot in terms of seconds ($s$).

**Influence of label rate on test accuracy and quality of compatibility matrix estimation.** Another interesting question under semi-supervised learning to study is the influence of label rates. Fig. 7 presents the CLP's test accuracy and the quality of compatibility matrix estimation ($\hat{\mathbf{H}}$) with different splittings. Specifically, the quality of $\hat{\mathbf{H}}$ is evaluated by $dist(\mathbf{H}, \hat{\mathbf{H}}) = \sqrt{\sum_{i=1}^{|\mathcal{Y}|} \sum_{j=1}^{|\mathcal{Y}|} \left([\mathbf{H}]_{ij} - [\hat{\mathbf{H}}]_{ij}\right)^2}$. It is not surprising to find that higher label rates lead to better performance and more accurate compatibility matrix estimation. Therefore, one of the future directions is to learn better compatibility matrix estimation according to prior predictions and graph structure.

**The number of parameters and execution time comparison.** Our approach often requires significantly fewer parameters than GNN models since only the base predictor has parameters to train, which is less than GNN models. Moreover, another gain is faster training time because we do not use the graph structure for our prior predictions, and the LP step is time-efficient (Gatterbauer, 2014; Gatterbauer et al., 2015). As an example, we plot the number of parameters vs test accuracy of CLP and H2GCN of two heterophily graphs, i.e., Brazil-Airports and Wisconsin, in Fig. 8. Note that H2GCN similarly contains an MLP component as a node feature encoder. We endow CLP and H2GCN with *Linear*, 2-layers and 3-layers MLP models as base predictors (feature encoder for H2GCN). The hidden dimensions of CLP and H2GCN are the same as the general settings. Each model's execution time (average value of 10 runs) under different settings is shown in Fig. 8. We observe that CLP achieves much better performance with orders of magnitude fewer parameters and execution time.

## 7 Conclusion

In this paper, we focused on the graph learning tasks with challenging heterophily settings. Motivated by an experimental investigation of existing models' performance, we proposed an approach that extends LP algorithm to heterophily settings by smoothing the prior predictions across neighbours weighted by the compatibility matrix. A theoretical analysis shows that CLP has a closed-form solution with mild conditions on an appropriate matrix and we can thus give a detailed explanation of when CLP will support convergence. Comprehensive experiments demonstrate the effectiveness and efficiency of our approach on real-world and synthetic graphs with different settings. In future work, we plan to investigate a better compatibility matrix estimation approach and generalise CLP to the heterophily setting of regression problems on graphs.

**Acknowledgments**

This work is supported by the Luxembourg National Research Fund through grant PRIDE15/10621687/SPsquared.

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
