# OpenReview forum: "Simplifying Node Classification on Heterophilous Graphs with Compatible Label Propagation"
_TMLR — Accepted by TMLR_

### Review · Reviewer_3KmR · 2022-07-01

**Summary Of Contributions:**

This work proposes a new method to perform node classification over heterophilic networks. The new method first uses features on each node with the label to train a simple model to predict the label. Then, use this model to predict pseudo labels of the rest nodes in the graph.  Further, use pseudo labels to estimate the compatibility matrix which is essentially the weight for the message passing to neighbors in different classes. Finally, combine pseudo labels with the compatibility matrix to perform a Pagerank-type diffusion to get the final prediction. Experiments on real and synthetic datasets demonstrate the good performance of the method.

**Broader Impact Concerns:**

I did not see any concerns about the ethical side of this work.

**Requested Changes:**

1. The results of real data experiments in all splitting settings should be provided.

2. Citation format is improper. Use \citep instead of \cite to add parentheses: e.g. xxx (Huang et al. 2021).

3. Consider doing extensive experiments on synthetic CSBMs as suggested in [1].


[1]  Adaptive Universal Generalized PageRank Graph Neural Network. ICLR 2021

**Strengths And Weaknesses:**

Strength:
1. The method sounds to be new for node classification over heterophilic networks.

2. The empirical performance looks good.

Weakness:
1. Although the method is new, the idea is not completely novel. It combines previous works, the C&S framework for homophilic networks [1] plus compatibility matrix-based node classification over heterophilic networks [2,3].

2. The method is ad-hoc. Previous work [2] gets the compatibility matrix-based node classification from belief propagation, while this work designs all components in a heuristic way, especially the design of the weighted adjacency matrix (Eq.(7)) /

3. Real dataset evaluation misses other splitting settings and only presents the medium splitting case.

4. The convergence analysis in 5.4 seems trivial to me, which is a standard result to analyze linear iterations.

[1] Combining Label Propagation and Simple Models Out-performs Graph Neural Networks. ICLR 2021
[2] Linearized and Single-Pass Belief Propagation. VLDB 2017
[3] Graph Neural Networks with Heterophily. AAAI 2021

---

> ### Author Response · Authors · 2022-07-29
> **Response to Reviewer 3KmR**
>
> We thank the reviewers for the detailed feedback, and we have taken the suggestions to improve the paper.
> Please, see the responses to the requested changes.
>
> > The results of real data experiments in all splitting settings should be provided.
>
> We provided supplement results, i.e., Table 4 and Table 5, into Appendix D to present the node classification results on real-world homophily and heterophily graphs, under $\textit{sparse}$ and $\textit{dense}$ splitting.
>
> > Citation format is improper. Use \textbackslash citep instead of \textbackslash cite to add parentheses: e.g. xxx (Huang et al. 2021).
>
> We have updated \cite to \citep through the paper.
>
> > Consider doing extensive experiments on synthetic CSBMs as suggested in [1].
>
> We included experiments on $19$ real-world datasets in addition to experiments with synthetic datasets, which we believe give strong evidence that our algorithm works well in practice.
> However, we will run experiments with more datasets, including CSBMs, in our future work.
>
> [1] Adaptive Universal Generalized PageRank Graph Neural Network. ICLR 2021

---

### Review · Reviewer_moVS · 2022-07-01

**Summary Of Contributions:**

This paper studies the effect of label propagation (LP) for transductive setting. It starts with the definition of homophily using homophily ratio. Then authors synthesize datasets with various homophily ratios and test MLP and GNNs on it. An empirical example illustrates that MLP is better in the lower homophily ratio case. Then motivated by this, authors design an LP algorithm by considering the compatibility. The empirical results can verify the effectiveness of CLP.


**Requested Changes:**

Some minor points:
1. In abstract, it seems that the following two sentences are redundant: “On a wide variety of benchmarks, …” and “Empirical evaluations demonstrate that …”. Can authors double-check this?
2. Authors can change the citation format (e.g., by modifying the natbib or link color). The current format is not friendly to readers.
3. In Sec 1, authors discussed the motivation, from homogeneous graph to heterogeneous graph. It would be better to movr the definition (i.e. the homophily ratio) in advance.
4. The CLP composes of three steps, yet there are existing works adopt the same pipeline, e.g. [1]. The main difference is on step (2): [1] calculates uncertainty for correction, and this work proposes a compatibility matrix for correction; the other two steps (base prediction and label smoothing) are the same. Authors may as well give credit to [1] at this place. Now authors only mention “steps 1 and 3 have been tried independently …”
5. The definition of homophily ratio of individual node (Eq 3) can be merged into Definition 1.
6. Some citations are wrong. For example, the citations for CPGNN are different in several places in the paper. Besides, the citation [2] in Sec 2 is H2GCN, not CPGNN. The authors should carefully check it.
7. According to the discussion in Sec 2 on the difference between H2GCN, authors should add it in Fig 1.


[1] Huang, Qian, et al. "Combining label propagation and simple models out-performs graph neural networks." arXiv preprint arXiv:2010.13993 (2020).
[2] Zhu, Jiong, et al. "Beyond homophily in graph neural networks: Current limitations and effective designs." Advances in Neural Information Processing Systems 33 (2020): 7793-7804.


**Strengths And Weaknesses:**

## Strengths:
1. The algorithm design part is reasonable, i.e., by adding compatibility into LP, though there are some issues on the motivation and analysis (in the Weaknesses).
2. The empirical results compare the latest LP and GNNs for heterophily.

## Weaknesses:
1. [The motivation example is confusing.] As mentioned by the authors, this compatibility measure was first introduced in a GNN work (the authors should fix the citations in Sec 2, as will be explained in Requested Changes). Later authors conduct an empirical study as a motivation to show that MLP is better than GNN in the low homophily ratio case. However, there seem to have several main issues on the logic here.
a. LP is not compared or discussed in Sec 4.
b. Now the conclusion the authors present is: because MLP is better than GNN (using compatibility)  in the low homophily case, we should  consider handling heterophily in LP (in the next section).
c. However, what I expect after reading Sec 4 is that, recent GNN models designed for heterophilous graphs do not outperform MLP in the low homophily case, so we should improve/design other GNNs that can avoid this.
d. Thus, combining the above three points, I don’t think the toy example is suitable here. Authors can consider adding more explanations or replacing it with another motivation example.

2. Later authors design an algorithm called CLP. It combines compatibility into LP. I have following questions here.
a. Why recent GNNs based on compatibility matrix can fail in this case, while LP with compatibility can work? The authors imply in Sec 4 that these recent GNNs may capture global information, but why this won’t happen for LP with compatibility (CLP)? Namely, both (GNN and LP) are using compatibility, why LP is better than GNN in capturing the “local” information? This is like the most fundamental question in this work; otherwise, this core idea of this paper is like: we empirically try LP with compatibility, and we observe that it works better than GNN with compatibility.
b. The authors cover some analysis on the CLP to prove its convergence. This is great. Just about the connection to question a, are authors implying that GNNs with compatibility fail to converge and CLP can achieve this? If so, authors can explicitly claim and prove it (or refer to existing work). I haven’t seen any theories on GNNs fail to converge in this case, so feel free to add the citations if authors have them in mind.
c. Besides, the analysis also provides the efficiency guarantee. The efficiency is only briefly discussed in Sec 6 (in OOM), which is not of the key focus of the current draft. If authors want to highlight the efficiency of CLP, then more experimental analysis is required. Otherwise, may consider moving the efficiency analysis to appendix.
d. I think the authors can consider adding more analysis on question a. The efficiency in the current draft does not seem to be the main focus, and authors may consider leaving more space for the more fundamental question.

3. The key recent GNNs are not sufficiently discussed in Sec 2, like GPRGNN, FAGCN, H2GCN.

4. The writing of this paper can be further improved. See minor points in Requested Changes.

---

> ### Author Response · Authors · 2022-07-29
> **Response to Reviewer moVS**
>
> We appreciate the reviewer’s time to provide us with the feedback that we take into account in our updated version. Please, see the responses to our questions below.
>
> > The motivation example is confusing.
>
> a. “LP is not compared or discussed in Sec 4” -> This is because LP does not work with node features.
>
> b. “we should improve/design other GNNs that can avoid this.” -> The focus of our work is not on the design of a new GNN model but on designing LP model that works well in the heterophily case because, as we show, C\&S model [1] does not work well in this case.
>
> > Analysis of CLP model
>
> a. “Why recent GNNs based on compatibility matrix can fail in this case, while LP with compatibility can work?” -> Analysing GNNs with compatibility matrix is beyond the scope of this paper. Some insights can be found in the original papers [2,3].
>
> b. “these recent GNNs may capture global information, but why this won’t happen for LP with compatibility (CLP)?” -> In this paper, we focused on the design of LP algorithm and showing that it works well in practice.
> We also provided additional analysis on the convergence of the algorithm. This provides an alternative to GNN that works well in practice.
> Analysing theoretical reasons for its success is kept for future work.
>
> c. “are authors implying that GNNs with compatibility fail to converge and CLP can achieve this?” -> no, we don’t claim this.
> In fact, as GNNs are parametric models trained by GD, we believe it has the same convergence guarantees as NNs.
>
> > The key recent GNNs are not sufficiently discussed in Sec 2, like GPRGNN, FAGCN, H2GCN.
>
> We discuss related work, like GPRGNN, FAGCN, H2GCN, in more detail in the updated version.
>
> > (Minor) In abstract, it seems that the following two sentences are redundant: “On a wide variety of benchmarks, …” and “Empirical evaluations demonstrate that …”. Can authors double-check this?
>
> We have revised the Abstract to remove redundant sentences.
>
> > (Minor) Authors can change the citation format (e.g., by modifying the natbib or link color). The current format is not friendly to readers.
>
> We have updated \cite to \citep and modified cite colour throughout the paper to make the paper easy to read.
>
> > (Minor) In Sec 1, authors discussed the motivation, from homogeneous graph to heterogeneous graph. It would be better to move the definition (i.e. the homophily ratio) in advance.
>
> We have moved the definition of the homophily ratio into Section 1.
>
> > (Minor) The CLP composes of three steps, yet there are existing works adopt the same pipeline, e.g. [1]. The main difference is on step (2): [1] calculates uncertainty for correction, and this work proposes a compatibility matrix for correction; the other two steps (base prediction and label smoothing) are the same. Authors may as well give credit to [1] at this place. Now authors only mention “steps 1 and 3 have been tried independently …”
>
> Thank you. We gave credit to [1] in the updated version.
>
> > (Minor) The definition of homophily ratio of individual node (Eq 3) can be merged into Definition 1.
>
> Since we have moved the definition of homophily ratio in Section 1, and Equation (3) is out if context of Section 1, therefore we keep it in Section 4.
>
> > (Minor) Some citations are wrong. For example, the citations for CPGNN are different in several places in the paper. Besides, the citation [2] in Sec 2 is H2GCN, not CPGNN. The authors should carefully check it.
>
> We have revised the mentioned incorrect citation and have carefully gone through other citations of the paper.
>
> > (Minor) According to the discussion in Sec 2 on the difference between H2GCN, authors should add it in Fig 1.
>
> We have the results of H2GCN in Figure 1.
>
> [1] Huang, Qian, et al. "Combining label propagation and simple models out-performs graph neural networks." arXiv preprint arXiv:2010.13993 (2020).
>
> [2] Zhu, Jiong, et al. "Beyond homophily in graph neural networks: Current limitations and effective designs." Advances in Neural Information Processing Systems 33 (2020): 7793-7804.
>
> [3] Bo et al., “Beyond Low-frequency Information in Graph Convolutional Networks”, 2021, AAAI

---

### Review · Reviewer_xGNs · 2022-07-15

**Summary Of Contributions:**

This paper proposed a new Graph Neural Network (GNN) using Label Propagation (LP) that was effective for prediction tasks on graphs with various heterophily.
First, this paper pointed out that existing GNNs did not have good prediction accuracies on nodes with high local heterophily, even though they were designed for high heterophily graphs.
Based on this finding, this paper proposed a new GNN model, Compatible Label Propagation (CLP). CLP estimated the compatibility matrix using node features and predicted node labels using compatible label propagation.
This paper applied CLP to semi-supervised node prediction tasks on the synthesis and real graphs and verified its effectiveness in terms of accuracy and implementation merits.

**Broader Impact Concerns:**

This paper does not have a Broader Impact Statement. I did not find any ethical concerns that would require a Broader Impact Statement.

**Requested Changes:**

If I understand correctly, the statement of Proposition 1 looks invalid. Therefore, I suggest re-examining it.

**Strengths And Weaknesses:**

Strengths

- Presentation is good.
- Proposed method is easy to implement.
- Claims are supported by extensive experiments.

Weaknesses

- Theoretical statements are seemingly mathematically invalid.


Soundness

- This paper claimed that in semi-supervised transductive learning on graphs, existing GNNs did not perform well on nodes with high local heterophily. Experiments in Section 4 empirically verified this claim.
- If I understand correctly, some mathematical statements appear to be invalid. Specifically, the Kronecker product $P\otimes Q$ is defined for the matrices $P$ and $Q$. However, since $A^F$ is an order-3 tensor, $I \otimes A^F$ in Proposition 1 eq. (8) is inconsistent.
- This paper claimed the following three points in the empirical aspect of the proposed method:
  - Good overall performance on various heterophily graphs.
  - Good accuracy on nodes with high local heterophily.
  - Efficient in terms of computational complexity and number of parameters.
These claims are numerically verified by experiments.

Relevance

The problem of performance degradation found in Section 4 is a critical issue of GNNs that few papers have addressed. Therefore, developing methods for solving it is a relevant topic for the TMLR community.

Novelty

As pointed out in this paper, existing studies such as Zhu et al. (2020) proposed models that utilized compatibility matrices. This paper explained the difference between proposed and existing methods in, e.g., Section 5.3. In addition, this paper numerically verified the effectiveness of the difference. Therefore, this paper has novelty and significance from this viewpoint.

Clarity

- The overall organization of the paper is good. Therefore, I can understand the paper without much difficulty.
- The introduction explained the line of research about heterophily GNNs and LP-based GNNs well and appropriately led us to the motivation of the proposed method.
- The explanation of the proposed method's ideas and methodology is good.

---

> ### Author Response · Authors · 2022-07-29
> **Response to Reviewer xGNs**
>
> > The statement of Proposition 1 looks invalid. Therefore, I suggest re-examining it.
>
> Thank you for your insightful reviews.
>
> We have updated Proposition 1 and its proof in Appendix B.

---

> > ### Comment · Reviewer_xGNs · 2022-08-11
> > **Answer to authors' response**
> >
> > I thank the authors for considering my comments and updating the draft. However, I still have questions about propositions. Therefore, I would suggest the authors to checking the statement and proof.
> >
> > If my understanding is correct, $\mathbf{I}$ is a matrix (=order-2 tensor) and $\mathbf{I} \odot \mathbf{A}^{\mathbf{F}}$ is an order-3 tensor. Therefore, $\mathbf{I} - \alpha \mathbf{I} \odot \mathbf{A}^{\mathbf{F}}$ is an invalid operation. In addition, It is not obvious how to define the inverse $\cdot^{-1}$ for an order-3 tensor.
> >
> > I also have several questions about the proof in the appendix. First, given the equation $[\hat{\mathbf{B}}]\_{:, k} = (1-\alpha) [\hat{\mathbf{D}}]\_{:, k} + \alpha \mathbf{A}^{\mathbf{F}}[\hat{\mathbf{B}}]\_{:, k}$, the vectorization operator is applied to it. However, it does not take any effect because both sides are already vectors. In addition, the transformation from eq. (20) to eq. (21) is difficult to interpret because the inverse operator for a rank-3 tensor is not obvious, as I mentioned above.
> >
> > Regarding Proposition 2,  we can only apply the spectral radius operator $\rho(\cdot)$ to a matrix, while $\mathbf{A}^{\mathbf{F}}$ is an order-3 tensor. Therefore it is difficult to interpret $\rho(\mathbf{A}^{\mathbf{F}})$.
> >
> > Additional comment: In Eq. (5), Definition of $\mathbf{Y}$ is missing. I guess this is the concatenation of one-hot encoded target labels.

---

> > > ### Author Response · Authors · 2022-08-11
> > > **Response to Reviewer xGNs - Part 2**
> > >
> > > > If my understanding is correct, $\mathbf{I}$ is a matrix (=order-2 tensor) and $\mathbf{I} \odot \mathbf{A}^{\mathbf{F}}$ is an order-3 tensor. Therefore, $\mathbf{I} - \alpha \mathbf{I} \odot \mathbf{A}^{\mathbf{F}}$ is an invalid operation. In addition, It is not obvious how to define the inverse $\cdot^{-1}$ for an order-3 tensor.
> > > I also have several questions about the proof in the appendix. First, given the equation $[\hat{\mathbf{B}}]_{:, k} = (1-\alpha) [\hat{\mathbf{D}}]_{:, k} + \alpha \mathbf{A}^{\mathbf{F}}[\hat{\mathbf{B}}]_{:, k}$, the vectorization operator is applied to it. However, it does not take any effect because both sides are already vectors. In addition, the transformation from eq. (20) to eq. (21) is difficult to interpret because the inverse operator for a rank-3 tensor is not obvious, as I mentioned above.
> > >
> > > We are sorry for this confusing part and thank the reviewer for the concrete question.
> > > We have revised Section 5.4 to clarify Proposition 1 and 2, and updated Appendix B to fix the corresponding proofs.
> > > Specifically, CLP iterative update node class label prediction of each class in turn (as shown in Equation (8)), therefore we present Proposition 1 and Proposition 2 for class $k$ and discuss their validity, respectively.
> > > Similarly, we can easily generalise these propositions to other class labels.
> > >
> > > > Regarding Proposition 2, we can only apply the spectral radius operator $\rho(\cdot)$ to a matrix, while $\mathbf{A}^{\mathbf{F}}$ is an order-3 tensor. Therefore it is difficult to interpret $\rho(\mathbf{A}^{\mathbf{F}})$.
> > >
> > > Thank you for the insightful comments. Following the revision of Proposition 1 (Section 5.4), we have updated Proposition 2 and its proof (Appendix B)
> > >
> > >
> > > > Additional comment: In Eq. (5), Definition of $\mathbf{Y}$ is missing. I guess this is the concatenation of one-hot encoded target labels.
> > >
> > > We thank the reviewer for this observation.
> > > The definition of $\mathbf{Y}$ was mistakenly deleted in the previous revision.
> > > We have revised Eq. (5) with the definition of $\mathbf{Y}$.

---

### Review · Reviewer_5knR · 2022-07-16

**Summary Of Contributions:**

The combination of label propagation and a simple shallow neural network can be used as an alternative of graph neural networks and achieve comparable performance on graphs with high homophily. However, it fails on heterophilous graphs. This paper generalizes LP algorithm to solve the heterophily issue by learning the class compatibility matrix and aggregating label predictions with LP algorithm weighted by the class compatibilities. It is demonstrated faster and performs well on graphs with various homophily levels.

**Broader Impact Concerns:**

Not applicable.

**Requested Changes:**

Weakness 1-3.

**Strengths And Weaknesses:**

Strength:

* A simple and efficient graph learning approach in heterophily settings.
* The find-grained analysis of performance on nodes with different local homophily levels is interesting and maybe inspiring.
* Seemingly good experimental performance with fast execution speed.

Weakness:

* Clarity. The definition of compatibility matrix in Equation (2) does not have enough explanation, and lacks of connection with Section 5.2.

* Notations. In (6), \hat{H} is derived from \hat{B}; in (7) F is derived from \hat{B} and \hat{H}; and then in (8) \{B} is further derived from \{F}? The notations are quite confusing here. I suppose the authors mean \hat{B}^0 in (6) and (7). If so, is A^F changed per layer, or computed only from the initial prediction and then fixed for all layers?

* Experiments. It is good to have three different split settings which may help explain the advantage of proposed methods in a certain setting. However, the experiments do not show clear performance difference in difference split settings in the synthetic data; while on real-world graphs, they even only use the medium setting, which is different from all previous papers. In my opinion, on real-world graphs they should either use the same setting as previous papers, or show results on all three different split settings. Otherwise, it is a little astonishing to see so many SOTA methods perform so badly compared to MLP, especially considering that they perform much better in their original split setting.

* What if we do not use a base network to learn the compatibility matrix H but only use the known ground truth labels (in the training part) to get it? This ablation study may be beneficial to show the advantage of learning the compatibility matrix.

---

> ### Author Response · Authors · 2022-07-29
> **Response to Reviewer 5knR**
>
> Thank you for your detailed suggestions and reviews.
>
> > Clarity. The definition of compatibility matrix in Equation (2) does not have enough explanation, and lacks of connection with Section 5.2.
>
> We revised Definition 2 and the explanation of the compatibility matrix before Definition 2 and changed relevant parts of Section 5.2.
>
> Particularly, Definition 2 provides one intuitive idea about the compatibility matrix that measures the fraction of outgoing edges from a node in class $i$ to a node in class $j$.
> The example of Appendix A gives an intuitive explanation of how $\mathbf{H}$ measures the variability of the homophily level. In Section 5.2, we empirically estimate a compatibility matrix $\hat{\mathbf{H}}$ by calculating the probability that a training node of one class is connected with a node of another class.
>
> > Notations. In (6), $\hat{H}$ is derived from $\hat{B}$; in (7) F is derived from $\hat{B}$ and $\hat{H}$; and then in (8) $\mathbf{B}$ is further derived from $\mathbf{F}$? The notations are quite confusing here. I suppose the authors mean $\hat{B}^0$ in (6) and (7). If so, is $A^F$ changed per layer, or computed only from the initial prediction and then fixed for all layers?
>
> Thank you for pointing out this confusing part.
> We have updated Equations (5) (6) and (7) and relevant parts of Section 5.2 and Section 5.3.
> We utilise $\hat{\mathbf{B}}^{(0)}$ to compute $\mathbf{A}^{\mathbf{F}}$ and $\mathbf{A}^{\mathbf{F}}$ will be fixed per layer of compatible label propagation process.
>
> > Experiments. In my opinion, on real-world graphs they should either use the same setting as previous papers, or show results on all three different split settings.
>
> We have added supplement results, i.e., Table 4 and Table 5, into Appendix D to present the node classification results on real-world homophily and heterophily graphs, under $\textit{sparse}$ and $\textit{dense}$ splittings.

---

### Author Response · Authors · 2022-08-01
**Summary of changes.**

Dear reviewers,

Thank you for the time to read our paper and provide us with valuable feedback. We made the changes according to your requests, which can be summarized as follows:

1. We corrected mathematical notations of compatibility matrix and equations.
2. We provided experiments with different splitting criteria in the real datasets.
3. We fixed the citation format and provided missing citations.
4. We made minor changes to make the paper even more clear.

The changes can be seen in the uploaded version of the paper and supplementary material. We hope these modifications address your concerns and we remain at your disposal.

---

### Decision · Action_Editors · 2022-09-01

**Recommendation:** Accept with minor revision

**Comment:**

This paper proposes a novel semi-supervised node classification method, called compatible label propagation (CLP).  The method first trains MLP on the training nodes based only on the node features, and computes the compatibility matrix by using the MLP predictions for the unlabelled nodes.  Then, label propagation is performed with the affinity matrix modified based on the compatibility matrix.

The proposed method is well-motivated, and shows SOTA performance.  Reviewers raised several major concerns on clarity, mathematical correctness, insufficient experiments, etc.  The authors have addressed most of the concerns and revised the paper accordingly.
Only the criticism on the limited novelty or technical contributions remains.  However, the difference from the existing methods is clearly described and marginal novelty cannot be the main reason for rejection according to the policy of TMLR.

Thanks to the reviewers' constructive comments and the authors' additional effort, the revised version is already in good shape except many minor flaws.  I only went through the revision quickly and didn't carefully check all the equations and sentences, but still I found a bunch of flaws.  I strongly recommend several authors (if this is a single author paper, ask two senior colleagues to proof-read) to check the whole manuscript and appendix very carefully.  The following are what I found.  I expect the authors will find much more flaws, all of which must be fixed before publication.

- Below Eq.(4), it is written \hat{D} = D^{(l)}, which is wrong.  It should read \hat{D} = Softmax(D^{(L)}), where l=L is the last layer.
- Above Eq.(5), "unknown nodes (V - T_V + T^v_V)".  Because each term is a set, set operators should be used: (V \setminus (T_V \setminus T^v_V)) or ((V \setminus T_V) \union T^v_V)).
- Below Eq.(8), What \| (double vertical lines) in \|_{k \in |Y|} means?  It's not defined.  Also, |Y| is the number of elements in the set Y, so k \in |Y| doesn't make sense.
- In Eq.(9), l is used for iteration counter, but it was used for the layer id in Eq.(4).  Use another character.
- Below Proposition 2, I don't understand what you mean by "Hence, one can use Frobenius or 1-induced norm to obtain sufficient convergence guarantees faster."?  Probably you should say "Hence, one can use Frobenius or 1-induced norm to efficiently check if the sufficient condition for convergence is satisfied."  Also, in "In our experiments, we found that CLP converges efficiently for all datasets." I don't understand what "efficient convergence" means.  Is there "inefficient convergence"?
- You can use Proposition 2 to bound the hyperparameter alpha for convergence guarantee.  Did you do this?  It's better to mention how you used Proposition 2 for your implementation.
- In the last paragraph of Section 5, "F is a function that guarantees that each message is a probability distribution."  I don't think this sentence defines F uniquely.  Can F be a constant distribution ignoring the argument?
- In Section 6.2, "We consider four different choices for ..." but I see only "sparse", "medium", and "dense".  What is the other one?
- In the same paragraph "in (Kipt & Welling, 2017)" should read "in Kipt & Welling, 2017" (use \citet here).
- In Table 2, what the green and purple highlightings mean (should be explained in the caption)?  Also better to apply statistical tests, e.g., Wilcoxon signed-rank test, and highlight the methods which are not significantly worse than the best method.  The same highlightings should be applied to Tables 4 and 5 in Appendix.


Minor requests (not mandatory but recommended):

- The CLP result should be included in Figure 1 already, and you can mention in Section 4 how good the proposed method is.  I suppose CLP behaves similarly to in Figure 5 but it's better to show real data results.  Also showing that the proposed CLP outperforms the other methods at this point will better motivate readers to read the subsequent sections.

- The proposed algorithm computes \hat{H} only once.  I'm curious what happens if you update \hat{H} after you get a better label prediction by LP than the prior prediction by MLP.  What happens if you iterate this process until convergence?  Possibilities are (1) the performance is degraded and it performs as other GNN approaches, (2) performance doesn't change much, or (3) performance improves.  I recommend the authors to investigate this point.  If the results are (1) or (2), it would be nice to give a single sentence explaining what happened.  If (3), it's nice to include the results in the paper with further improved performance.

- I'd say "the number of parameters" instead of "parameter number".

- I'd say "existing methods", "SOTA methods", "baseline methods", instead of "Competing methods"

---

> ### Author Response · Authors · 2022-09-29
> **Response to Reviewer Action Editors**
>
> We appreciate the time and efforts you and the reviewers have dedicated to providing valuable feedback, affirmations, and insightful comments, which helped us improve the manuscript. Special thanks to the editor for the meticulous review process and organisation. We have been able to incorporate the changes to reflect the concerns provided by the editor for the camera-ready version. Below is a point-by-point response to the editor’s comments and concerns.
>
> > Below Eq.(4), it is written \hat{D} = D^{(l)}, which is wrong. It should read \hat{D} = Softmax(D^{(L)}), where l=L is the last layer.
>
> Thank you for noticing this point, we have revised it accordingly.
>
> > Above Eq.(5), "unknown nodes (V - T_V + T^v_V)". Because each term is a set, set operators should be used: (V \setminus (T_V \setminus T^v_V)) or ((V \setminus T_V) \union T^v_V)).
>
> Thank you for noticing this point, we have revised it accordingly.
>
> > Below Eq.(8), What | (double vertical lines) in |_{k \in |Y|} means? It's not defined. Also, |Y| is the number of elements in the set Y, so k \in |Y| doesn't make sense.
>
> Thank you for noticing this point, we have revised it accordingly.
>
> > In Eq.(9), l is used for iteration counter, but it was used for the layer id in Eq.(4). Use another character.
>
> Thank you for noticing this point, we have revised it accordingly.
>
> > Below Proposition 2, I don't understand what you mean by "Hence, one can use Frobenius or 1-induced norm to obtain sufficient convergence guarantees faster."? Probably you should say "Hence, one can use Frobenius or 1-induced norm to efficiently check if the sufficient condition for convergence is satisfied." Also, in "In our experiments, we found that CLP converges efficiently for all datasets." I don't understand what "efficient convergence" means. Is there "inefficient convergence"?
>
> Thank you for noticing this point, we have revised it accordingly.
>
> > You can use Proposition 2 to bound the hyperparameter alpha for convergence guarantee. Did you do this? It's better to mention how you used Proposition 2 for your implementation.
>
> Thank you for noticing this point, we added a description at the end of Appendix E to explain how our hyperparameter settings satisfy the convergence guarantee.
>
> > In the last paragraph of Section 5, "F is a function that guarantees that each message is a probability distribution." I don't think this sentence defines F uniquely. Can F be a constant distribution ignoring the argument?
>
> Thank you for noticing this point, we have revised it accordingly.
>
> > In Section 6.2, "We consider four different choices for ..." but I see only "sparse", "medium", and "dense". What is the other one?
>
> Thank you for noticing this point, we have revised it accordingly.
>
> > In the same paragraph "in (Kipt & Welling, 2017)" should read "in Kipt & Welling, 2017" (use \citet here).
>
> Thank you for noticing this point, we have revised it accordingly.
>
> > In Table 2, what the green and purple highlightings mean (should be explained in the caption)? Also better to apply statistical tests, e.g., Wilcoxon signed-rank test, and highlight the methods which are not significantly worse than the best method. The same highlightings should be applied to Tables 4 and 5 in Appendix.
>
> Thank you for your comments. We have updated the captions of the experimental result tables. And we added a Rank column in experimental result tables to show the overall rank of each model in all datasets. It can demonstrate the general performance of each model.
>
> > I'd say "the number of parameters" instead of "parameter number".
>
> Thank you for noticing this point, we have revised it accordingly.
>
> > I'd say "existing methods", "SOTA methods", "baseline methods", instead of "Competing methods"
>
> Thank you for noticing this point, we have revised it accordingly.